# Sensing minimal self in sentences involving the speaker

Ryoko Uno[1], Shu Imaizumi [2]*

**1** Division of Language and Culture Studies, Institute of Engineering, Tokyo University of Agriculture and Technology, Tokyo, Japan, **2** Institute for Education and Human Development, Ochanomizu University, Tokyo, Japan

* imaizumi.shu@ocha.ac.jp

## Abstract

The role of language in the narrative self is well-known, but does it also affect the minimal self? We investigated whether variations in sentence structure affect the speaker's sense of the minimal self. Previous research has examined how third-person subject expressions influence interpretations of agency, particularly in causal contexts. We examined whether expressing the first-person subject's involvement in causation or perception events influences the speaker's sense of agency and ownership, key components of the minimal self. Participants completed an online experiment using psychological rating scales to evaluate Japanese sentences with varying degrees of speaker involvement, as if they had uttered them. Each sentence varied in whether and how it encoded causation or perception. We analyzed Japanese data since, in addition to perceiver-prominent sentences (e.g., *[watashi ga] hoshi o mita*, "I saw a star"), Japanese has perceiver-stimulus-prominent sentences (e.g., *[watashi ni] hoshi ga mieta*, "A star was visible to me"/ "I could see a star"), which also foreground the first-person perceiver in a double subject construction. We found that the speaker's sense of agency was significantly lower in sentences where either causation or perception was absent (e.g., *hoshi ga deteita*, "A star was out") compared to sentences where either was present. Agency was also significantly higher for perceiver-prominent sentences than for perceiver-stimulus-prominent sentences. Regarding ownership, it was also significantly higher for perceiver-prominent sentences. Whether ownership was significantly higher for perceiver-stimulus-prominent sentences than for those lacking causation or perception varied with the perceived stimulus. These results suggest that variations in linguistic structure can distinctly impact the senses of agency and ownership. In cognitive linguistics, certain sentence structures are analyzed to reflect how subjectively the speaker is construed. Our findings suggest that differences in agency and ownership provide an empirical basis for this argument, grounded in embodied experience.

**Data availability statement:** The dataset used for the analysis can be found at the Open Science Framework (https://osf.io/sz3p8/).

**Funding:** This study was supported by KAKENHI (21K12603 and 25K15839 to RU and 20H04094, 20K20144, and 23K11785 to SI) from the Japan Society for the Promotion of Science (https://www.jsps.go.jp/english/). The funder had no role in study design, data collection and analysis, decision to publish, or preparation of the manuscript.

## Introduction

People love talking about themselves. Behavioral and neuroimaging studies show that people talk a lot about themselves [1] and talking about oneself is essentially rewarding for humans [2]. This use of language makes our experience coherent over extended periods of time. The notion of a narrative self is studied in various fields—including philosophy, psychology, and neuroscience [3]—and the relationship between this type of self and narrative language is almost evident. But, what about the use of a sentence whose main purpose is not to describe the speaker's coherent self, such as "I saw a flower" or "The flower is in bloom"?

While narrative self is a "self" that is extended over time, the "self" experienced as an immediate subject of experience in a moment is called minimal self [3]. It is argued that the use of the first-person pronoun in a self-referring manner is related to the minimal self [4–6]. This predicts that when a sentence reports an event that involves the first person, then it might reflect the sense of the minimal self.

Psychological studies of self assume that the minimal self has two aspects [3,7,8]. The first is the sense of agency, which refers to the feeling that "I am the one who causes the action." The second is the sense of ownership, which refers to the sense that "I am the one who is undergoing the experience." Although these two senses can correlate [9], they are generally considered conceptually distinct [3,7]. Several studies have demonstrated that the two senses can be dissociated behaviorally [10–12] and even neurally [13]. Recently, not only how people sense these aspects of the minimal self but also how they can be generated by using tools or virtual reality that extend the subject's body is being intensively studied (e.g., [14–16]).

In cognitive linguistics, an approach that examines the embodiment of language, there are numerous discussions on the relationship between sentence structure and the speaker's self-recognition. Most notably, Langacker [17,18] argues that the degree of the speaker's involvement in linguistic expressions is related to the extent to which the speaker is construed subjectively or objectively. In this argument, the concepts used to describe the speaker's self-recognition are original to cognitive grammar, and how they relate to those in the psychological study of the self remains unclear. In this paper, we aim to lay the foundation for connecting rich linguistic observations of how the speaker is involved in the events described in sentences with the psychological study of the self. Building on previous attempts to measure the senses of agency and ownership using self-report psychological scales [12,19,20], we conducted an experiment to examine how the speaker's use of sentences involving the first person as an agent in causation or as an experiencer of perception correlates with the sense of the minimal self.

### Agency and ownership in language expressions

Departing from the self-recognition or first-person perspective, there is a great deal of research that inquires about the relationship between agency and linguistic expressions. In contrast, how the sense of ownership is reflected in language is not studied much. This gap leaves open questions regarding how linguistic structures might shape a speaker's perception of ownership in their experiences.

Agency is reflected in language at least in two dimensions: performance and encoding [21]. When we make people do things by using language, we are performing agency. If we say "Break the vase!" the listener would try to break a vase. In this paper, we do not deal with this dimension; we limit our argument to the encoded agency in language. The following two sentences can be used to describe the same situation but differ in terms of encoding agency. Sentence (1) is agentive, but sentence (2) is not.

| (1) She broke a vase. |
|---|
| (2) A vase was broken. |

Using agentive and non-agentive pairs, such as (1) and (2), Fausey et al. [22] analyzed how agency is verbalized and the effect it has on cognition. They compared how the language difference in describing agency affects the difference in the cognition of agency by contrasting Japanese and English. In Japanese, in many cases, the contrast between agentive and non-agentive sentences can be expressed with transitive and intransitive verbs that share a root. For example, below, sentence (3) is agentive and sentence (4) is non-agentive. For the transcription of the Japanese examples, we used the Hepburn system of Romanization. For the word-to-word translations, we used the abbreviations ACC for accusative, DAT for dative, NOM for nominative, PAST for past tense, and PROG for progressive aspect.

| (3) |
|---|
| *kanojo ga kabin o watta.* |
| she NOM vase ACC break-PAST |
| "She broke a vase." |
| (4) |
| *kabin ga wareta.* |
| vase NOM break-PAST |
| "A vase broke." |

An analysis of the above sentences reveals that the linguistic expression of agency correlates with the sense of agency not only across languages (e.g., English and Japanese) but also within a single language (e.g., across sentence types in English). In other studies, language is used as a measure for the sense of agency. For example, Oren et al. [23] observed linguistic expressions that describe agency in third-person scenarios used by the subjects to measure their sense of agency; their results revealed that the sense of agency is mitigated when the subject has an obsessive-compulsive tendency; this, in turn, can relate to decreased action selection and loss of autonomy.

Most of these studies focus on the detection of agency in others. Our paper investigates the sense of self and, thus, the main focus is on the agency of the first person—the speaker themselves. We are interested in the contrast between the non-agentive sentences in sentences (2) and (4) and the agentive sentences with first-person pronouns as a subject in sentences (5) and (6). In Japanese, it is possible—and often more natural—to omit the first-person subject, as demonstrated in sentence (6).

| (5) I broke the vase. |
|---|
| (6) |
| (*watashi ga*) *kabin o watta.* |
| (I NOM) vase ACC break-PAST |
| "I broke a vase." |

The sense of ownership of one's own body or experience is also regarded as an integral aspect of the sense of minimal self. It is a sense that conveys that "I am the one who is undergoing the experience." While agency in language is studied well in causal events, as we have seen in examples (1) to (4), and can be applied to the analysis of the agency of the speaker as in (5) and (6), for the sense of ownership, we have to choose which events expressed by language are adequate for the study of ownership in language. There are several possibilities in this regard. We could continue to analyze causal events, contrasting

the case where the first person pronoun (i.e., the speaker) is encoded as an agent or a patient. Another possibility is focusing on the descriptions of emotion or perception by the speaker, positioning the speaker as an experiencer. In this paper, we pursue the latter possibility and analyze sentences related to visual perception using Japanese expressions.

Visual perception can be expressed in various forms. It can be described as an intentional action, such as "look at," as shown in (7), as well as an unintentional action, such as "see" in (8).

| |
|---|
| (7) I looked at a flower. |
| (8) I saw a flower. |

Further, it can be described as even less intentional, not as an action of "I" but a property of the flower when the target of perception is the subject and the adjective "visible" is used, as in (9).

| |
|---|
| (9) A flower was visible to me. |

Finally, we can entirely erase the existence of the speaker in the sentence and describe the same situation objectively, as in (10) or (11).

| |
|---|
| (10) A flower was visible. |
| (11) A flower was in bloom. |

In this paper, we compare the following three sentences in Japanese to investigate the sense of ownership and the sense of agency in using language. Sentence (12) is the Japanese equivalent of sentences (7) or (8). In other words, the verb *miru* can be used to express both intentional seeing and unintentional seeing. In sentence (12), omitting the first-person subject *watashi* ("I") is more natural in Japanese.

Sentence (13) is the Japanese equivalent of sentence (9). One notable difference is that sentence (13) can be analyzed as a double subject construction: in sentence (13), not only "a flower" but also "I" can be regarded as a subject. (To illustrate this point, the translation "I could see a flower" is added.) The noun *hana* ("a flower") bears the nominative case marker *-ga*, and although *watashi* ("I") takes the dative marker *-ni*, it can still be considered a subject, since the use of *mieru* in its honorific form elevates the perceiver—not the object of perception.

In sentence (13), it is more natural to omit the first-person *watashi* ("I"). Note that the verb stems in sentences (12) and (13) share the same root. Furthermore, sentence (14) is the Japanese equivalent of sentence (11).

| |
|---|
| (12) |
| (*watashi ga*) *hana o mita*. |
| (I NOM) flower ACC see-PAST |
| "I {looked at/saw} a flower." |
| (13) |
| (*watashi ni*) *hana ga mieta*. |
| (I DAT) flower NOM visible-PAST |
| "A flower was visible to me./ I could see a flower." |
| (14) |
| *hana ga saiteita*. |
| flower NOM bloom-PROG-PAST |
| "A flower was in bloom." |

Sentences (4), (6), (12), (13), and (14) are examples of five sentence types used in our experiment.

## Materials and methods

This study was not pre-registered. The dataset used for the analysis can be found at the Open Science Framework (https://osf.io/sz3p8/).

## Participants

One hundred twenty-one students participated in this experiment for partial course credit. The participants were recruited from an introductory linguistics class at the Tokyo University of Agriculture and Technology from May 28th to 31st, 2021. Two participants whose mother tongue was not Japanese were excluded from the analysis. The remaining 119 participants were analyzed (95 men, 23 women, one unknown; mean age of 22.6 years, age range 21–33 years, $SD = 1.2$). Our limited resources were the reason for this sample size.

All participants provided electronic informed consent prior to the experiment, as it was conducted online rather than in person. The first page of the experimental website presented information about the purpose of the study and ethical considerations. Participants who read and understood this information and agreed to participate in the experiment gave their consent by checking the "I agree to participate in the study" box on the screen. Their responses were recorded along with other data collected during the experiment. This study, along with the consent procedure, was approved by the Ethics Committee of the Tokyo University of Agriculture and Technology (approval number: 201202−0263).

## Materials

Ten target sentences were presented to participants. These sentences were designed to systematically vary along two dimensions: whether the event was with or without causation (Table 1) and whether it was with or without perception (Table 2). Tables 1 and 2 summarize the ten target sentences according to this two-dimensional framework. We set two scenes for each dimension: scenes "breaking a vase" and "growing a flower" for causation, and "seeing a flower" and "seeing a star" for perception.

To avoid terminological confusion with the concept of *agency*, which we empirically measured in this study, we refrain from using the term *agent*—a semantic role label derived from the same root. Instead, we use the term *actor* to refer to the participant who performs the action in causation events, and *undergoer* to refer to the target of the action. In perception events, we use the term *perceiver* to refer to the experiencer, and *stimulus* to refer to the target of perception. Since our purpose is to measure the sense of self, all the actors or perceivers in these sentences are in the first person—that is, the speaker.

We then classified the sentences based on syntactic prominence—specifically, which participants function as the grammatical subject(s). Since subjects typically reflect the participant with the highest cognitive salience, this classification helps identify which element is conceptually foregrounded in each sentence type. The three categories are:

(i) Actor-/Perceiver-prominent: only the actor or perceiver is realized as the subject.

(ii) Perceiver-stimulus-prominent: both the perceiver and the stimulus appear in subject-like positions. This type occurs only in perception events. Compared to single prominence in (i), in double prominence the perceiver may be less fully foregrounded, but is still prominent.

(iii) Undergoer-/Stimulus-only: only the undergoer or stimulus is realized as the subject; the actor or perceiver is absent.

**Table 1. Target sentences with and without causation.**

|  | Actor-prominent sentence (with causation) | Undergoer-only sentence (without causation) |
|---|---|---|
| Breaking a vase | *kabin o watta.* "I broke a vase." | *kabin ga kowareta.* "A vase broke." |
| Growing a flower | *hana o sodateta.* "I grew a flower." | *hana ga sodatta.* "A flower grew." |

The target sentences that were actually presented to participants were in Japanese. The English translation follows the Japanese expression.

 

**Table 2. Target sentences with and without perception.**

| | Perceiver-prominent sentence (with perception) | Perceiver-stimulus-prominent sentence (with perception) | Stimulus-only sentence (without perception) |
|---|---|---|---|
| Seeing a flower | *hana o mita.* "I {looked at/saw} a flower." | *hana ga mieta.* "A flower was visible to me./ I could see a flower." | *hana ga saiteita.* "A flower was in bloom." |
| Seeing a star | *hoshi o mita.* "I {looked at/saw} a star." | *hoshi ga mieta.* "A star was visible to me./ I could see a star." | *hoshi ga deteita.* "A star was out." |

The target sentences that were actually presented to participants were in Japanese. The English translation follows the Japanese expression.

We selected these sentence types because (i) and (ii) express causation or perception, whereas (iii) does not. Each type can offer an alternative description of the same event, with (i) and (ii) explicitly marking the speaker's involvement, and (iii) omitting it.

Passive constructions could theoretically serve as candidates for "undergoer-prominent (with causation/perception)" sentences. However, we do not include them in this study, as there are no natural or acceptable passive counterparts to type (i) sentences in Japanese. The following examples in sentences (15) and (16) are not acceptable:

| (15) |
|---|
| ?? *kabin ga watashi ni warareta.* |
| "A vase is broken by me." |

| (16) |
|---|
| ?? *hana ga watashi ni mirareta.* |
| "A flower was seen by me." |

Each target sentence was embedded in an instruction in Japanese, which can be translated in the following manner: "Suppose the sentence 'I broke a vase.' is the one you produced. Choose the extent to which each item is applicable to this sentence." (Original Japanese: *"kabin o watta." to iu bun o, anata ga nobeta bun da to shimasu. kono bun ni tsuite kakukoumoku ga dorehodo atehamaru ka o erande kudasai*.) The instruction was intended to have participants read the sentence as their own and imagine the scene described in the target sentence.

Participants responded to each of seven items (Table 3) using a seven-point Likert scale ranging from 1 "Not applicable at all (*mattaku atehamaranai*)" to 7 "Extremely applicable (*hijoo ni atehamaru*)" for each target sentence. Items 1 and 2 were included to assess a sense of agency, which is experienced by the speaker, while items 3 and 4 were included to assess a sense of ownership. Items 5, 6, and 7 were included because they are thought to be related to imagery, affinity, and familiarity, respectively, and thus serve as control measures that are irrelevant to senses of agency and ownership. These items were developed with reference to a previous scale designed to measure the senses of agency and ownership in everyday experiences [20]. Therefore, the items used in the present study are likely to be easy for participants to understand and respond to.

## Procedures

Participants completed the experiment via SurveyMonkey (http://www.surveymonkey.com), using their own computer. Instructions and target sentences were set to be displayed in black font on white background. The font type and size varied based on the participant's web browser. Each target sentence and its rating scale items were presented on separate web pages. The order of target sentences was randomized for each participant. The order of items on the rating scale was randomized for each target sentence. Finally, participants were asked to report their age, gender, and mother tongue. The median of the completion time was 367 s (interquartile range of 292–497).

**Table 3. Rating scale items.**

| Scale | # | Item |
|---|---|---|
| Agency | 1 | *jibun ga kono bamen o hikiokoshita kanji ga suru.*<br>"I feel as if I caused this scene." |
| | 2 | *jibun ga kono bamen o okosu koto o ito shita kanji ga suru.*<br>"I feel as if I intended to cause this scene." |
| Ownership | 3 | *jibun ga kono bamen o keiken shita kanji ga suru.*<br>"I feel as if I experienced this scene." |
| | 4 | *kono bamen de erareru keiken o jibun no mono toshite kanjiru.*<br>"I feel that I own the experience gained from this scene." |
| Imagery | 5 | *kono bamen o senmei ni souzou suru koto ga dekiru.*<br>"I can vividly imagine this scene." |
| Affinity | 6 | *kono bamen o konomashiku kanjiru.*<br>"I like this scene." |
| Familiarity | 7 | *kono bamen o natsukashiku kanjiru.*<br>"I feel nostalgic about this scene." |

The English translation follows the Japanese expression

## Results

### Analysis procedures

Since scores for items 1 and 2 across target sentences were strongly and positively correlated (Spearman's $\rho$ = 0.712, 95% confidence interval [0.683, 0.739], $p$ < 0.001), the average score of these two items served as a composite variable called *Agency*. The other five items exhibited weak or null correlations with items 1 and 2 ($\rho$s = −0.013–0.442). Similarly, since items 3 and 4 were moderately correlated ($\rho$ = 0.658 [0.624, 0.689], $p$ < 0.001), their average served as a composite variable called *Ownership*. The other items exhibited weak correlations with items 3 and 4 ($\rho$s = 0.301–0.492). Control items 5–7 showed weak correlations with each other ($\rho$s = 0.205–0.483), thereby suggesting that they should remain separate variables.

The rating scores for each of the four scenes were separately submitted to repeated measure analysis of variance with two within-participant factors: Scale (Agency, Ownership, Imagery, Affinity, Familiarity) and Sentence. The Sentence factor included two levels (actor-prominent, undergoer-only) for the scenes describing breaking a vase and growing a flower, and three levels (perceiver-prominent, perceiver-stimulus-prominent, stimulus-only) for the scenes describing seeing a star and seeing a flower. Given that non-normal dependent variables do not substantially affect the type I error and power of $F$ statistics in repeated measure analysis of variance [24], we performed this analysis with non-normal or ordinal rating scores. Moreover, the Greenhouse-Geisser correction was applied to the degree of freedom when the assumption of sphericity was violated based on Mauchly's test. Post-hoc tests were performed with Holm's correction. The statistical significance level was set at $p$ < 0.05. Data analysis was performed using JASP 0.19.2 [25].

### Comparison between actor-prominent and undergoer-only

For the rating scores on target sentences describing the "breaking a vase" scene (Fig 1), we found significant main effects of Scale ($F$(3.0, 359.8) = 259.32, $p$ < 0.001, $\eta^2_p$ = 0.687) and Sentence ($F$(1, 118) = 114.93, $p$ < 0.001, $\eta^2_p$ = 0.493) and their interaction ($F$(3.7, 433.4) = 50.87, $p$ < 0.001, $\eta^2_p$ = 0.301). In addition, significant differences between two sentences were found for the Agency and Ownership scales ($t$s > 6.98, $p$s < 0.001) but not for the others ($t$s < 1.98, $p$s > 0.050). Further, the effect size of the difference in Agency between two sentences (Cohen's $d$ = 1.54 [1.03, 2.04]) was much larger than that in Ownership ($d$ = 0.77 [0.37, 1.18]). For descriptive purposes, we report the effect size ($d$) of 0.17 [−0.15, 0.50] for Imagery, 0.05 [−0.12, 0.21] for Affinity, and 0.16 [−0.11, 0.44] for Familiarity.

In the "growing a flower" scene (Fig 2), there were significant main effects of Scale ($F(3.5, 408.1) = 38.50$, $p < 0.001$, $\eta^2_p = 0.246$) and Sentence ($F(1, 118) = 60.70$, $p < 0.001$, $\eta^2_p = 0.340$) and their interaction ($F(3.5, 413.5) = 39.81$, $p < 0.001$, $\eta^2_p = 0.252$). Significant differences between two sentences were found for the Agency, Ownership, and Familiarity ($ts > 2.42$, $ps < 0.017$) but not for the others ($ts < 1.73$, $ps > 0.087$). The difference in Agency between two sentences ($d = 1.34$ [0.85, 1.83]) was larger than that in Ownership ($d = 0.88$ [0.48, 1.27]). The effect sizes were 0.17 [−0.16, 0.51] for Imagery, 0.03 [−0.24, 0.31] for Affinity, and 0.28 [−0.11, 0.67] for Familiarity.

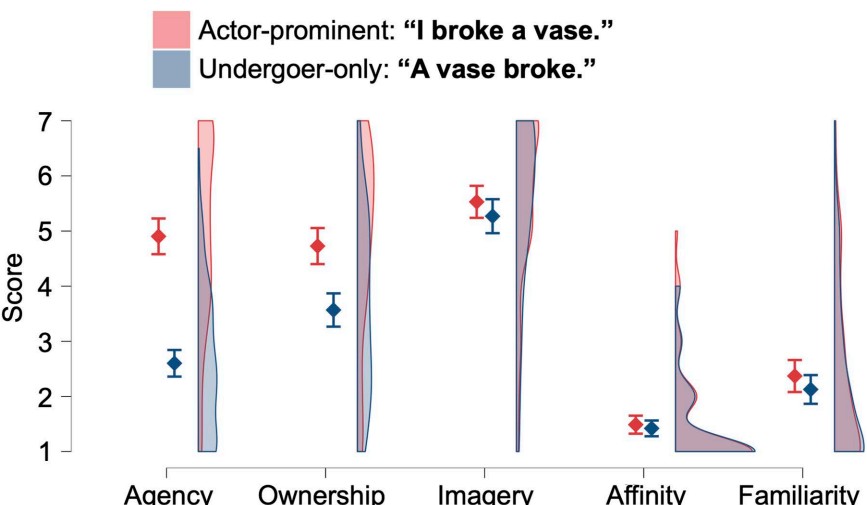

**Fig 1. Rating score for the actor-prominent and undergoer-only sentences describing scenes where a vase was broken.** Error bars represent 95% confidence intervals.

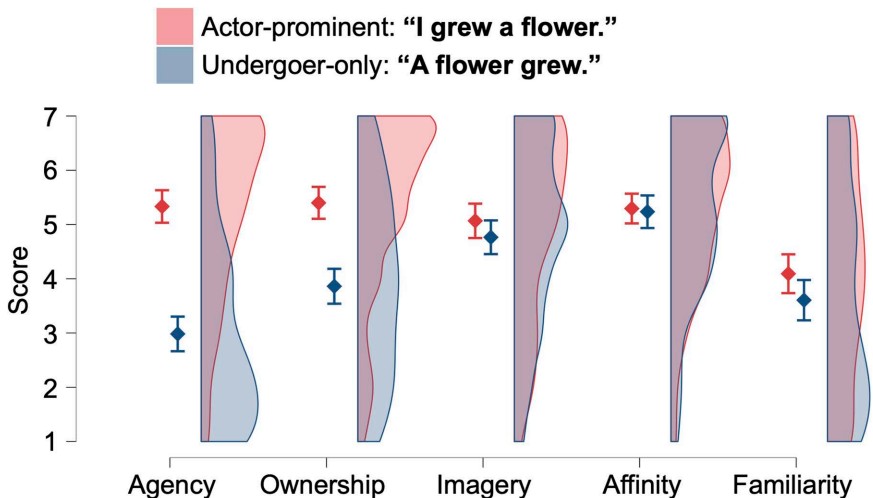

**Fig 2. Rating score for the actor-prominent and undergoer-only sentences describing scenes where a flower was grown.** Error bars represent 95% confidence intervals.

## Comparison of sentence types by perceiver prominence

For the rating scores on target sentences describing the "seeing a flower" scene (Fig 3), we found significant main effects of Scale ($F(3.7, 437.8) = 112.23$, $p < 0.001$, $\eta^2_p = 0.487$) and Sentence ($F(2, 236) = 28.65$, $p < 0.001$, $\eta^2_p = 0.195$) and their interaction ($F(6.4, 754.3) = 33.93$, $p < 0.001$, $\eta^2_p = 0.223$). The simple main effects of Sentence were found for all scales ($F$s > 5.18, $p$s < 0.006). Post-hoc tests revealed significant differences between sentences for all rating scales (for details, see Fig 3 and Table 4). Based on visual inspection, the differences among three sentences were steeper for Agency than for Ownership. Supporting this, the effect sizes of the differences in Agency between the perceiver-prominent and perceiver-stimulus-prominent sentences and between the perceiver-prominent and stimulus-only sentences were $d$s of 1.09 and 1.49, respectively, while those for Ownership were much smaller ($d$s = 0.40 and 0.52, respectively).

In the "seeing a star" scene (Fig 4), there were significant main effects of Scale ($F(3.6, 423.5) = 120.67$, $p < 0.001$, $\eta^2_p = 0.506$) and Sentence ($F(1.9, 224.6) = 44.91$, $p < 0.001$, $\eta^2_p = 0.276$) and their interaction ($F(6.2, 730.8) = 23.58$,

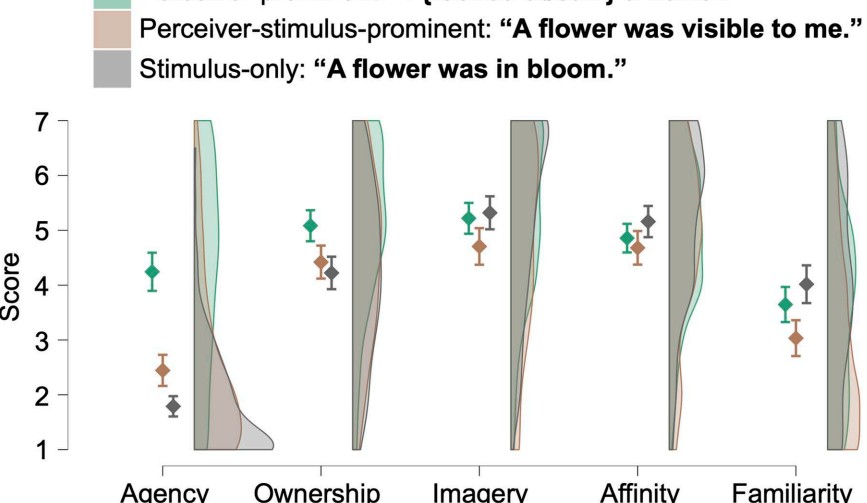

**Fig 3. Rating score for sentences describing scenes where a flower was seen.** Error bars represent 95% confidence intervals.

**Table 4. Effect sizes of the differences between target sentences describing the seeing-a-flower scene.**

| | "I {looked at/saw} a flower." and "A flower was visible to me. / I could see a flower." | "I {looked at/saw} a flower." and "A flower was in bloom." | "A flower was visible to me. / I could see a flower." and "A flower was in bloom." |
|---|---|---|---|
| Agency | $d = 1.09$ [0.56, 1.62] **$p < 0.001$** | $d = 1.49$ [0.93, 2.04] **$p < 0.001$** | $d = 0.40$ [0.07, 0.72] **$p < 0.001$** |
| Ownership | $d = 0.40$ [0.06, 0.75] **$p < 0.001$** | $d = 0.52$ [0.15, 0.89] **$p < 0.001$** | $d = 0.12$ [−0.22, 0.46] $p = 0.211$ |
| Imagery | $d = 0.31$ [−0.09, 0.71] **$p = 0.010$** | $d = -0.06$ [−0.43, 0.31] $p = 0.553$ | $d = -0.37$ [−0.72, −0.02] **$p < 0.001$** |
| Affinity | $d = 0.11$ [−0.26, 0.47] $p = 0.290$ | $d = -0.18$ [−0.46, 0.09] **$p = 0.032$** | $d = -0.29$ [−0.64, 0.06] **$p = 0.009$** |
| Familiarity | $d = 0.37$ [−0.03, 0.77] **$p = 0.002$** | $d = -0.22$ [−0.54, 0.10] **$p = 0.012$** | $d = -0.60$ [−1.02, −0.17] **$p < 0.001$** |

Brackets indicate 95% confidence intervals. Statistically significant values ($p < 0.05$, Holm-corrected) are given in bold.

$p < 0.001$, $\eta^2_p = 0.167$). Simple main effects of Sentence were found for all scales ($Fs > 4.55$, $ps < 0.012$) but not for the Imagery ($F = 2.14$, $p = 0.120$). Post-hoc tests revealed significant differences between sentences for all but Imagery scales (Fig 4 and Table 5). The effect sizes of the differences in Agency between the perceiver-prominent and perceiver-stimulus-prominent sentences and between the perceiver-prominent and stimulus-only sentences were $d$s of 0.98 and 1.62, respectively, while those for Ownership were much smaller ($d$s = 0.27 and 0.65, respectively).

## Discussion

This study aimed to explore how linguistic structures influence the speaker's experience of agency and ownership, which are two components of the minimal self, even in sentences that do not explicitly narrate personal experiences.

To begin with, as shown in Figs 1 and 2, we compared actor-prominent (with causation) and undergoer-only (without causation) sentences with the aim of examining how the speaker's sense of agency is influenced by different sentence types. The sense of agency is higher in actor-prominent sentences and lower in undergoer-only sentences. Similarly, the

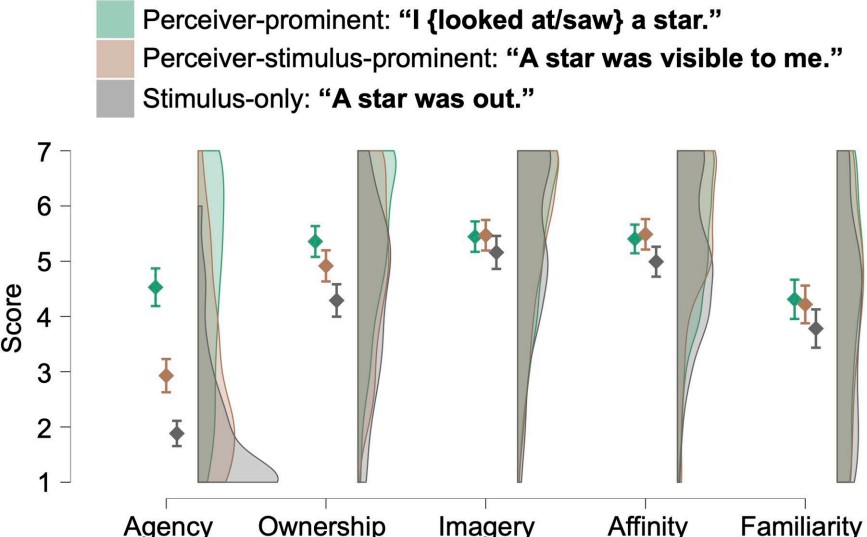

**Fig 4. Rating score for sentences describing scenes where a star was seen.** Error bars represent 95% confidence intervals.

**Table 5. Effect sizes of the differences between target sentences describing the seeing-a-star scene.**

|  | "I {looked at/saw} a star." and "A star was visible to me. / I could see a star." | "I {looked at/saw} a star." and "A star was out." | "A star was visible to me. / I could see a star." and "A star was out." |
|---|---|---|---|
| Agency | $d = 0.98$ [0.50, 1.46] **$p < 0.001$** | $d = 1.62$ [1.03, 2.20] **$p < 0.001$** | $d = 0.64$ [0.27, 1.00] **$p < 0.001$** |
| Ownership | $d = 0.27$ [−0.07, 0.61] **$p = 0.004$** | $d = 0.65$ [0.25, 1.05] **$p < 0.001$** | $d = 0.38$ [0.05, 0.71] **$p < 0.001$** |
| Imagery | $d = -0.02$ [−0.32, 0.29] $p = 0.856$ | $d = 0.18$ [−0.22, 0.57] $p = 0.255$ | $d = 0.19$ [−0.21, 0.59] $p = 0.255$ |
| Affinity | $d = -0.05$ [−0.33, 0.22] $p = 0.502$ | $d = 0.25$ [−0.09, 0.59] **$p = 0.016$** | $d = 0.30$ [−0.02, 0.62] **$p = 0.002$** |
| Familiarity | $d = 0.06$ [−0.34, 0.45] $p = 0.606$ | $d = 0.32$ [−0.13, 0.77] **$p = 0.030$** | $d = 0.27$ [−0.14, 0.67] **$p = 0.035$** |

Brackets indicate 95% confidence intervals. Statistically significant values ($p < 0.05$, Holm-corrected) are given in bold.

sense of ownership is also higher in actor-prominent sentences and lower in undergoer-only sentences. Additionally, we observed that while both senses exhibit lower rates in undergoer-only sentences compared to actor-prominent sentences, the decrease is more pronounced for agency than for ownership.

Our findings demonstrate that the two sentence types—actor-prominent and undergoer-only—affected not only the sense of agency but also the sense of ownership. Two possible explanations may account for this result. First, it could be attributed to the observation made in previous studies [8,26] that the sense of agency induces a sense of body ownership or the correlation between the sense of agency and sense of body ownership [10]. Second, it is also possible, although not mutually exclusive with the first explanation, that what are referred to as "agentive sentences" (which are called actor-prominent sentences in this paper) not only describes the agency of the subject but also incorporates the subject's ownership of bodily and perceptual experiences. In other words, when we break an object as an agent, we inevitably experience how we broke the object.

Next, as shown in Figs 3 and 4, we compared three sentence types—perceiver-prominent (with perception), perceiver-stimulus-prominent (with perception), and stimulus-only (without perception)—to examine their impact on the speaker's sense of ownership. While the distinction between agentive and non-agentive sentences corresponds to the difference between actor-prominent and undergoer-only sentences, the analysis becomes more complex when the sentient or perceiver aspect is considered. It must be noted that not only perceiver-prominent but also perceiver-stimulus-prominent sentences involve first-person perception. Nevertheless, the perceiver's prominence or foregroundedness may be reduced in double-prominence constructions compared to single-prominence ones, where the perceiver is the sole grammatical focus.

Our results revealed a decline in the sense of agency across these sentence types in the order of perceiver-prominent, perceiver-stimulus-prominent, and stimulus-only. The pattern for the sense of ownership was more nuanced. In the flower scene, the sense of ownership was significantly higher for perceiver-prominent sentences than for both perceiver-stimulus-prominent and stimulus-only sentences. However, in the star scene, the ownership was significantly higher for both perceiver-prominent and perceiver-stimulus-prominent sentences than for stimulus-only ones, although there was significant but small difference between perceiver-prominent and perceiver-stimulus-prominent sentences. Similar to the comparison between actor-prominent and undergoer-only sentences, the decline in agency was more pronounced than that in ownership.

It is important to note that, although a perception-event sentence lacks an agent in terms of semantic role, the presence of a subject experiencing perception can still result in a high agency rating. This observation, coupled with the finding that so-called agentive sentences (i.e., actor-prominent sentences) evoke not only a strong sense of agency but also a sense of ownership, leads us to speculate that the semantic role "agent" is not an exact equivalent of the sense of agency that is a part of the minimal self.

We must also consider the dual nature of perceiver-stimulus-prominent sentences. In the star scene, ownership ratings for this sentence type were nearly as high as those for perceiver-prominent sentences, with a significant but small difference. In contrast, in the flower scene, ownership ratings dropped to the level observed for stimulus-only sentences. We propose that this contrast may arise from whether expressions of perceptual availability—such as *visible to me* or the Japanese verb *mieru*—typically assume the existence of the perceived stimulus, except in some cases such as pareidolia [27], where one sees a face in the stains on the ceiling while knowing that none is actually there. In the star scene, the perceiver-stimulus-prominent sentence tends to presuppose that the star exists and highlights the act of perception (e.g., "The sky was clear, so I was able to see the star"). In some cases, it may also serve to report the discovery of the star's presence through perception. In the flower scene, however, the same sentence type more naturally evokes a context in which the existence of the flower is not presupposed. It would be pragmatically odd to point to a clearly visible flower in front of both the speaker and the hearer and use this structure to assert its existence. As a result, in the flower scene, the perceiver-stimulus-prominent sentence is more likely to be interpreted similarly to a stimulus-only sentence, which simply

reports the presence or state of the object. These results suggest that variation in ownership may reflect whether the sentence foregrounds the perceiver's experiential access or merely indicates the object's presence, although further research is needed to determine whether this interpretation is valid. This interpretation aligns with our experimental findings, particularly in the flower scene, where ownership ratings for perceiver-stimulus-prominent sentences were comparable to those for stimulus-only sentences.

Another aspect worth discussing is the relatively higher sense of ownership compared to agency in undergoer-/stimulus-only sentences. This suggests that even a mere description of the environment—without explicit reference to the first person—can still evoke a sense of ownership, though not agency. This observation raises the possibility that ownership may arise from sentences that describe a scene but implicitly presuppose a perceiver.

We believe that the present study contributes to advancing the theoretical discussion of subjectivity in the construal of the speaker in cognitive linguistics. This issue has long been explored by scholars such as Langacker [17,18], who argues that while the speaker typically occupies a subjective vantage point as the conceptualizer, this status may shift when the speaker is also a participant in the event being conceptualized. Depending on the degree and manner of linguistic encoding, the speaker may be construed more or less subjectively. A classic example of this is the contrast between the following two sentences:

| (17) |
| --- |
| a. There is snow all around me. |
| b. There is snow all around. |

Langacker [18] explains that in sentence (17a), the speaker explicitly refers to themselves, adopting an external perspective on their own situation. In contrast, sentence (17b) omits the first-person reference and instead evokes the scene from an internally grounded perspective. While the speaker remains the conceptualizer in both cases, the construal in sentence (17b) is more subjective, as the speaker is not explicitly mentioned.

Ikegami [28] builds on this idea and interprets the subjective construal of the speaker as arising when the speaker functions as the origin of perception. In such cases, the speaker's presence is implicitly assumed, and the omission of a first-person subject is natural. He points to Japanese expressions such as emotional adjective constructions (18) and perception-based sentences (19) as analogous cases. These constructions typically assume a first-person experiencer, who is often omitted and cannot be easily replaced by a second- or third-person subject. Sentences (18b, c) and (19b, c) are all unnatural in Japanese:

| (18) | |
| --- | --- |
| a. | (*watashi wa*) *ureshii.* |
| | "I am happy." |
| b. | ?? *anata wa ureshii.* |
| | "You are happy." |
| c. | ?? *kanojo wa ureshii.* |
| | "She is happy." |
| (19) | |
| a. | (*watashi ni*) *hoshi ga mieta.* |
| | "A star was visible to me." |
| b. | ?? *anata ni hoshi ga mieta.* |
| | "A star was visible to you." |
| c. | ?? *kare ni hoshi ga mieta.* |
| | "A star was visible to him." |

Based on such Japanese examples, Ikegami [28] argues that the subjective construal of the speaker, as exemplified in sentence (17b), is not merely a matter of whether the first-person subject is overtly expressed or omitted. Rather, it

reflects a broader and more complex phenomenon concerning how the speaker's presence as the origin of perception is linguistically established and constrained in relation to the scene.

In the current paper, we analyzed sentences like (19a) as perceiver–stimulus-prominent sentences, and what we observed may be connected to the "immersed perspective" discussed by Langacker and Ikegami. While our interpretation aligns with their core arguments, we acknowledge that both frameworks have faced critical scrutiny. For instance, differences in how Langacker and Ikegami conceptualize the subjective construal of the speaker have been noted [29]. It has also been pointed out that Langacker's theoretical treatment of examples such as sentences (17a, b) [17,30–32] has changed across publications [33], suggesting that the theoretical account of speaker subjectivity is still evolving.

These considerations highlight the value of experimental approaches that can offer fine-grained, empirically grounded insights into how the speaker is construed subjectively. Our findings provide a possible empirical correlate of such subjective construal. In the star scene, perceiver–stimulus-prominent sentences show a specific combination—relatively low agency and relatively high ownership—which may correspond to the type of speaker construal that Langacker and Ikegami describe as subjective. Interestingly, even in other conditions where agency was clearly low (e.g., perceiver–stimulus-prominent sentences in the flower scene or stimulus-only sentences for both the star and flower scenes), ownership did not decline as steeply. While these differences were not always statistically significant, they suggest a gradient rather than a binary phenomenon.

If this hypothesis holds, it implies that the sense of "being inside the scene" or "experiencing from within," as described in previous linguistic theories, may be observed not just as a categorical distinction but as a matter of degree—quantifiable through patterns in ownership and agency. This offers a novel way to connect linguistic intuition with measurable psychological experience and opens a new avenue for integrating linguistic and cognitive models of subjectivity.

Our findings can be robust as the order and carry-over effects on responses were eliminated by full randomization of the target sentence and response scale orders. It is unlikely that demand characteristics and response biases led to apparent effects of target sentences on senses of agency and ownership since we found these effects are quite weak or null for control items (e.g., Imagery). Nevertheless, we acknowledge three limitations. First, we investigated the speaker's minimal self as reflected in linguistic expressions, based on participants' evaluations. However, in reality, participants evaluated the target sentences as if they themselves had spoken them. Therefore, it is possible that our experiment did not measure the speakers' own minimal self, but rather a minimal self inferred by others. This makes the relationship between the minimal self and its linguistic expression opaque. Given that our findings rely on such indirect evidence for minimal self in sentences, it would be beneficial to evaluate participants' own utterances, for example, in agentive and non-agentive situations, and to compare narratives by individuals with normal and abnormal experiences of minimal self (e.g., [23,34]). Second, our results were obtained from a university student sample of which the majority was men. Third, our conclusions were drawn from responses to four scenes described in ten target sentences. Therefore, replication and extension studies with various sentences should be conducted in samples with broader age and educational status and more balanced gender ratio.

Finally, our findings, in conjunction with previous studies that reveal how speakers' minimal selves influence and interact with their speech and narratives, will bring a new perspective to future research on minimal and narrative selves. It has been suggested that characteristic language use reflects an altered sense of agency in individuals with psychiatric disorders; for example, people with a high obsessive-compulsive tendency exhibit omission of agents from the sentences they produce [23] and those with high schizotypal personality and schizophrenia display a lack of coherence [35] and fewer descriptions of agency in their self-narratives [34]. These findings suggest a possibility that the presence of a subject and the voice of verbs in a sentence may also reflect the speaker's own minimal self and, in contrast, may influence the minimal self perceived from sentences, as in our findings. In nature, the minimal self has been believed to be generated through perceptual experiences and motor actions [3,7,36] and transiently modulated or altered by sensorimotor and multisensory signals [8,10]. The present findings suggest that the minimal self could also be modulated by language

comprehension and production and, thus, add a linguistic factor to the explanation of the mechanism of the minimal self. Language is a key component for the self narrated and represented in one's mind. Therefore, it can be hypothesized that language use constructs the minimal and narrative selves and even mediates between them; future studies should investigate this possibility.

## Acknowledgments

We would like to thank Sayaka Hasegawa, Masayuki Ishizuka, Yoshiki Nishimura, and Taichi Tanaka for the helpful comments on this manuscript.

## Author contributions

**Conceptualization:** Ryoko Uno, Shu Imaizumi.

**Formal analysis:** Shu Imaizumi.

**Funding acquisition:** Ryoko Uno, Shu Imaizumi.

**Investigation:** Ryoko Uno, Shu Imaizumi.

**Visualization:** Shu Imaizumi.

**Writing – original draft:** Ryoko Uno, Shu Imaizumi.

**Writing – review & editing:** Ryoko Uno, Shu Imaizumi.

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
