## [Decision Letter · Decision Letter 0]

27 Feb 2025

PONE-D-25-00361
Sensing minimal self in a sentence that involves the speaker
PLOS ONE

Dear Dr. Imaizumi,

Thank you for submitting your manuscript to PLOS ONE. After careful consideration, we feel that it has merit but does not fully meet PLOS ONE’s publication criteria as it currently stands. Therefore, we invite you to submit a revised version of the manuscript that addresses the points raised during the review process.

While the research question of how language affects the minimal self is novel, both reviewers raise questions concerning whether the methods employed can address them. Moreover, both reviewers take issue with aspects of the theoretical framework governing this paper, as well as interpretation of the findings. I am recommending that the author be allowed to revise the paper to address these and other concerns raised by the reviewers. Should they choose to do so, the paper will be returned to the original reviewers so that they can evaluate the extent to which their concerns are addressed.

We look forward to receiving your revised manuscript.

Kind regards,

Laura Morett

Academic Editor

PLOS ONE

Journal Requirements:

Reviewers' comments:

Reviewer's Responses to Questions

**Comments to the Author**

1. Is the manuscript technically sound, and do the data support the conclusions?

Reviewer #1: Partly

Reviewer #2: Partly

2. Has the statistical analysis been performed appropriately and rigorously? 

Reviewer #1: I Don't Know

Reviewer #2: I Don't Know

3. Have the authors made all data underlying the findings in their manuscript fully available?

Reviewer #1: Yes

Reviewer #2: Yes

4. Is the manuscript presented in an intelligible fashion and written in standard English?

Reviewer #1: Yes

Reviewer #2: Yes

5. Review Comments to the Author

Reviewer #1: The study investigates how different types of sentences affect one's minimal sense of self, specifically examining sense of agency and ownership. For the sense of agency, the findings reveal a clear pattern: objective sentences resulted in low levels of agency, while action and perception type sentences result in high levels. For the sense of ownership, the relation found was more complicated and depended on context (the content of the sentence). For some sentences, the sense of ownership was significantly higher in "sentient-action" sentences compared to both "sentient-situation" and "non-agentive-non-sentient" sentences, while for others, the first two types of sentences showed high levels with only objective sentences scoring low. The authors explore what these findings imply regarding the links between semantic and psychological notions of agent.

While the paper's central claim about linguistic expressions impacting the minimal sense of self is not particularly surprising, it offers an intriguing reversal of the typical research direction, which usually examines how minimal sense of self contributes to narratives rather than vice versa. The findings regarding sense of ownership are noteworthy, though the proposed explanations require further substantiation. Despite the complexity of the subject matter and some imprecise conceptual distinctions, the paper's approach is commendable. However, there are several concerns to address, with one major issue standing out.

A significant concern regarding the methodology is whether the measures are actually capturing the intended phenomenon. When subjects are given sentences such as "I broke the vase", telling them that these are their sentences will not necessarily engage the minimal self in the way the authors envisage. Evidently, the subjects didn't break a vase and, hence, when they are given such a sentence and are asked to, say, rate a statement such as "I feel as if I caused this scene", they might engage in some (implicit) imaginative narrativising. They might do something along the lines of: "I am someone who says X. If I were X, what would I feel about causing the scene?" If that is the case, responses to the measures in the study wouldn't reliably report sense of (minimal) self. While this doesn't necessarily exclude the involvement of the minimal self (maybe via some process discussed by the literature on the simulation theory of mind), it's not the only possible explanation for what's being measured. Alternative approaches might yield more reliable results, such as employing non-linguistic measures combined with sentences that participants genuinely own, or perhaps utilizing virtual reality to enhance immersion. The authors should address these methodological concerns or at least acknowledge that such alternative explanations need to be excluded through different experimental paradigms in future research.

Now to the smaller comments:

1. "we aim [...] to introduce a method for measuring subjectivity." (l40-41)

- Given the above worry, I'm not sure the authors succeed at this.

2. "However, as Clark [12, 13] has repeatedly indicated, even without technology, humans can extend themselves with language." (l61-63)

- I am not sure the paper's subject matter concerns extended cognition. It is true that some of the research on minimal selfhood makes use of notions from that literature, but evidently we can also study the minimal (unextended) self. I think this is what the authors do in the present study.

3. The authors conceptual scheme (introduced in the 'Materials' sections) could use some clarification. First, the terms "agentive action" and "sentient action" aren't the most illuminating though I am unsure how to improve them. More generally, the authors' division of the stimulus sentences into "action" and "perception" instances is somewhat confusing. For instance, it seems to me that "A flower was visible to me" is an instance of perception in the same sense as it is an instance of action. Neither action nor perception require intentionality according to the authors (which, in itself, is a bit unorthodox), and what's happening in sentient situation sentences seems simply to be that the agent is the object rather than the subject of the sentence. Why would that remove the action component but not the perception component? Moreover, "The sentient action sentence describes a scene in which the speaker engages in an action to intentionally have a perception." (l205) is contradicted by the inclusion of "I saw a flower" into sentient actions. I am aware that this isn't a philosophy paper and, hence, maybe not quite the same level of conceptual busywork is needed, but I invite the authors to try and find a more elucidating/intuitive conceptual scheme.

4. Why are sentences of the type "a vase was broken (by me)" missing? They would, in my view, be akin to sentient situation sentences but of the action variety. This also links to the above worry (3) regarding sentient situation sentences.

5. "The observed effects on the first-person sense of agency in our experiment align with the findings of Fausey et al. [20], who investigated the recognition of agency in third-person contexts. However, their framework did not address the scaling of the sense of ownership, as this concept is applicable only in first-person contexts." (l331-334)

- Nitpick: It doesn't seem plausible to me that Fausey et al. didn't study the sense of ownership because they only studied third-person contexts. Couldn't they have e.g. prompted for rankings of statements such as "I feel like she experienced the situation" or "I think she feels like she experienced the situation", which would determine whether an agent attributes ownership to another person.

6. "This distance may suggest that the vision in the star scene is more private, whereas in the flower scene, it is more public, as anyone can see the flower" (l368 - 370)

- I am not convinced by this explanation, given that anyone can see stars too (if not more so, given their visibility from various vantage points). Because of this I'm also not convinced that the proposed further research will help settle the issue.

7. I am not sure I fully understand the discussion in the paragraph in lines 430 - 438, and I think other readers too might appreciate a little more explanation. What exactly is the claim? Does 'subjective construal of the speaker' here refer to the production of sentences of a certain type or the instantiation of certain mental states? Additionally, is this a causal claim (as implied by 'occur') and, if so, what exactly is causing what?

Reviewer #2: I find some difficulties in the presentation of the theoretical framework within which the experiment is conducted. First, the sense of the minimal self is usually considered prior to language; Gallagher 2000 for example, includes some reflections about the minimal self of a robot, and also we can claim that pre-linguistic children and other mammals have a sense of the minimal self. In this sense, the assertion that “language plays a role in creating the sense of the minimal self” is incorrect. May be language reflects this sense, but does not create it.

Secondly, this minimal self is constituted by the sense of agency and the sense of ownership, but these are not two elements that can be tested independently, because whenever there is a statement about an action, the sense of ownership is also involved in addition to the sense of agency, if the sense of minimal self is present. That’s why offering different linguistic expressions to test the sense of agency and the sense of ownership is debatable. Both senses should be present in order to state that the sense of the minimal sense is involved.

I do not understand the following sentence “An analysis of the above sentences reveals that the linguistic expression of agency is correlated with the sense of agency not only between languages (i.e., English and Japanese) but also within a

language (i.e., English)” (114 - 115)

The explanation proposed (365 - 370) regarding the sense of ownership in the example of the star and the flower does not seem convincing to me. The distance seems irrelevant, and I cannot understand why the perception of a star is “nor private” and the perception of a flower “more public”.

In fact, the problem might be that the expressions that are used in the scale to measure agency and ownership, (at least in the English version I do not know if it is the same in Japanese) are not expressions that we would use in our everyday life. To my knowledge, no one would say, without philosophical training, that they feel to “own” an experience, or to be the owner of a scene; the same probably goes for the expression "intended to cause." That is to say, the scale proposed to measure the senses of agency and ownership seems artificial and not ecological to me, introducing technical terminology that does not correspond to the usual ways of thinking about our own experience. In that sense I am not sure that they can measure anything at all.

In the design of the task that the subjects must perform, I find the proposal that the sentences should be imagined as spoken by them problematic, as well as the task of imagining the scene describe by that sentence. It seems to me that there are too many layers of cognitive processes involved in this task that make the relationship between the minimal self and its linguistic expression somewhat opaque, making the result of the paper inconclusive.

Finally, if we look at the abstract, it seems excessive to think that this experimental method can clarify what is involved in “subjectivity”, and I do not see how the results of this experiment can establish any connection between the narrative self and the minimal self, given that the narrative self includes a huge amount of elements, among them episodic memory, which is not mentioned in this text. The abstract promises much more that the paper does.

6. PLOS authors have the option to publish the peer review history of their article (what does this mean?). If published, this will include your full peer review and any attached files.

Reviewer #1: **Yes: **Julian Hauser

Reviewer #2: **Yes: **Diana Ines Pérez

---

## [Author Response · Author response to Decision Letter 1]

5 Jun 2025

[Authors] We sincerely thank Reviewer #1 and Reviewer #2 for their constructive comments on our manuscript. Please find our responses to each comment below.

Reviewer #1:

The study investigates how different types of sentences affect one's minimal sense of self, specifically examining sense of agency and ownership. For the sense of agency, the findings reveal a clear pattern: objective sentences resulted in low levels of agency, while action and perception type sentences result in high levels. For the sense of ownership, the relation found was more complicated and depended on context (the content of the sentence). For some sentences, the sense of ownership was significantly higher in "sentient-action" sentences compared to both "sentient-situation" and "non-agentive-non-sentient" sentences, while for others, the first two types of sentences showed high levels with only objective sentences scoring low. The authors explore what these findings imply regarding the links between semantic and psychological notions of agent.

While the paper's central claim about linguistic expressions impacting the minimal sense of self is not particularly surprising, it offers an intriguing reversal of the typical research direction, which usually examines how minimal sense of self contributes to narratives rather than vice versa. The findings regarding sense of ownership are noteworthy, though the proposed explanations require further substantiation. Despite the complexity of the subject matter and some imprecise conceptual distinctions, the paper's approach is commendable. However, there are several concerns to address, with one major issue standing out.

A significant concern regarding the methodology is whether the measures are actually capturing the intended phenomenon. When subjects are given sentences such as "I broke the vase", telling them that these are their sentences will not necessarily engage the minimal self in the way the authors envisage. Evidently, the subjects didn't break a vase and, hence, when they are given such a sentence and are asked to, say, rate a statement such as "I feel as if I caused this scene", they might engage in some (implicit) imaginative narrativising. They might do something along the lines of: "I am someone who says X. If I were X, what would I feel about causing the scene?" If that is the case, responses to the measures in the study wouldn't reliably report sense of (minimal) self. While this doesn't necessarily exclude the involvement of the minimal self (maybe via some process discussed by the literature on the simulation theory of mind), it's not the only possible explanation for what's being measured. Alternative approaches might yield more reliable results, such as employing non-linguistic measures combined with sentences that participants genuinely own, or perhaps utilizing virtual reality to enhance immersion. The authors should address these methodological concerns or at least acknowledge that such alternative explanations need to be excluded through different experimental paradigms in future research.

[Authors] As the reviewer pointed out, the present study merely provides indirect evidence for the relationship between a speaker’s minimal self and linguistic expression, since participants rated the sentences by inferring the minimal self experienced by others (i.e., speakers of the target sentences). We have acknowledged this limitation as follows, while also suggesting alternative measurement approaches that could more directly examine participants’ own minimal self.

Lines 513-521: First, we investigated the speaker’s minimal self as reflected in linguistic expressions, based on participants’ evaluations. However, in reality, participants evaluated the target sentences as if they themselves had spoken them. Therefore, it is possible that our experiment did not measure the speakers’ own minimal self, but rather a minimal self inferred by others. This makes the relationship between the minimal self and its linguistic expression opaque. Given that our findings rely on such indirect evidence for minimal self in sentences, it would be beneficial to evaluate participants’ own utterances, for example, in agentive and non-agentive situations, and to compare narratives by individuals with normal and abnormal experiences of minimal self (e.g., [23, 34]).

Now to the smaller comments:

1. "we aim [...] to introduce a method for measuring subjectivity." (l40-41)

- Given the above worry, I'm not sure the authors succeed at this.

[Authors] I clarified that our aim is not to measure subjectivity in general, but specifically the subjectivity of the speaker when they are involved in the described event. Here are the revised sentences:

Lines 42-45: In cognitive linguistics, certain sentence structures are analyzed to reflect how subjectively the speaker is construed. Our findings suggest that differences in agency and ownership provide an empirical basis for this argument, grounded in embodied experience.

2. "However, as Clark [12, 13] has repeatedly indicated, even without technology, humans can extend themselves with language." (l61-63)

- I am not sure the paper's subject matter concerns extended cognition. It is true that some of the research on minimal selfhood makes use of notions from that literature, but evidently we can also study the minimal (unextended) self. I think this is what the authors do in the present study.

[Authors] We agree that the focus of our study is on the minimal self and that invoking extended cognition may have introduced unnecessary conceptual ambiguity. To avoid this confusion, we have removed the entire paragraph that referred to Clark’s work, including the related citations.

3. The authors conceptual scheme (introduced in the 'Materials' sections) could use some clarification. First, the terms "agentive action" and "sentient action" aren't the most illuminating though I am unsure how to improve them. More generally, the authors' division of the stimulus sentences into "action" and "perception" instances is somewhat confusing. For instance, it seems to me that "A flower was visible to me" is an instance of perception in the same sense as it is an instance of action. Neither action nor perception require intentionality according to the authors (which, in itself, is a bit unorthodox), and what's happening in sentient situation sentences seems simply to be that the agent is the object rather than the subject of the sentence. Why would that remove the action component but not the perception component? Moreover, "The sentient action sentence describes a scene in which the speaker engages in an action to intentionally have a perception." (l205) is contradicted by the inclusion of "I saw a flower" into sentient actions. I am aware that this isn't a philosophy paper and, hence, maybe not quite the same level of conceptual busywork is needed, but I invite the authors to try and find a more elucidating/intuitive conceptual scheme.

[Authors] We appreciate your point that our use of the terms “agentive action” and “sentient action” may not be the most intuitive. In light of this, we have revised our terminology in the manuscript to more clearly reflect our two key event types: causation and perception events. As explained in the Materials section (lines 215-248), our new classification is based on syntactic prominence, specifically regarding which participants appear as subject(s).

Lines 215-248: Ten target sentences were presented to participants. These sentences were designed to systematically vary along two dimensions: whether the event was with or without causation (Table 1) and whether it was with or without perception (Table 2). Tables 1 and 2 summarize the ten target sentences according to this two-dimensional framework. We set two scenes for each dimension: scenes “breaking a vase” and “growing a flower” for causation, and “seeing a flower” and “seeing a star” for perception.

 ---Tables 1 and 2 inserted---

To avoid terminological confusion with the concept of agency, which we empirically measured in this study, we refrain from using the term agent—a semantic role label derived from the same root. Instead, we use the term actor to refer to the participant who performs the action in causation events, and undergoer to refer to the target of the action. In perception events, we use the term perceiver to refer to the experiencer, and stimulus to refer to the target of perception. Since our purpose is to measure the sense of self, all the actors or perceivers in these sentences are in the first person—that is, the speaker.

We then classified the sentences based on syntactic prominence—specifically, which participants function as the grammatical subject(s). Since subjects typically reflect the participant with the highest cognitive salience, this classification helps identify which element is conceptually foregrounded in each sentence type. The three categories are:

(i) Actor-/Perceiver-prominent : only the actor or perceiver is realized as the subject.

(ii) Perceiver-stimulus-prominent: both the perceiver and the stimulus appear in subject-like positions. This type occurs only in perception events. Compared to single prominence in (i), in double prominence the perceiver may be less fully foregrounded, but is still prominent.

(iii) Undergoer-/Stimulus-only: only the undergoer or stimulus is realized as the subject; the actor or perceiver is absent.

We selected these sentence types because (i) and (ii) express causation or perception, whereas (iii) does not. Each type can offer an alternative description of the same event, with (i) and (ii) explicitly marking the speaker’s involvement, and (iii) omitting it.

[Authors] We also added an explanation of the double subject construction in perceiver-stimulus-prominent sentences, highlighting how both the stimulus and the perceiver can function as subjects in Japanese, as follows:

Line 166-171: Sentence (13) is the Japanese equivalent of sentence (9). One notable difference is that sentence (13) can be analyzed as a double subject construction: in sentence (13), not only “a flower” but also “I” can be regarded as a subject. (To illustrate this point, the translation “I could see a flower” is added.) The noun hana (“a flower”) bears the nominative case marker -ga, and although watashi (“I”) takes the dative marker -ni, it can still be considered a subject, since the use of mieru in its honorific form elevates the perceiver—not the object of perception.

4. Why are sentences of the type "a vase was broken (by me)" missing? They would, in my view, be akin to sentient situation sentences but of the action variety. This also links to the above worry (3) regarding sentient situation sentences.

[Authors] We added an explanation to answer this point in the manuscript (lines 249–260). Passive constructions could theoretically serve as candidates for “undergoer-/stimulus-prominent (with causation/perception)” sentences. However, we do not include them in this study, as there are no natural or acceptable passive counterparts for “actor-/perceiver-prominent sentences” with first-person subjects in Japanese (e.g., ?? kabin ga watashi ni warareta “A vase was broken by me”).

Lines 249-260: Passive constructions could theoretically serve as candidates for “undergoer-prominent (with causation/perception)” sentences. However, we do not include them in this study, as there are no natural or acceptable passive counterparts to type (i) sentences in Japanese. The following examples in sentences (15) and (16) are not acceptable:

(15)

?? kabin ga watashi ni warareta.

“A vase is broken by me.”

(16)

?? hana ga watashi ni mirareta.

“A star was seen by me.”

5. "The observed effects on the first-person sense of agency in our experiment align with the findings of Fausey et al. [20], who investigated the recognition of agency in third-person contexts. However, their framework did not address the scaling of the sense of ownership, as this concept is applicable only in first-person contexts." (l331-334)

- Nitpick: It doesn't seem plausible to me that Fausey et al. didn't study the sense of ownership because they only studied third-person contexts. Couldn't they have e.g. prompted for rankings of statements such as "I feel like she experienced the situation" or "I think she feels like she experienced the situation", which would determine whether an agent attributes ownership to another person.

[Authors] We agree that the original sentence was misleading. We revised it to remove the reference to Fausey’s work. The revised version is as follows:

Lines 384-386: Our findings demonstrate that the two sentence types—actor-prominent and undergoer-only—affected not only the sense of agency but also the sense of ownership. Two possible explanations may account for this result.

6. "This distance may suggest that the vision in the star scene is more private, whereas in the flower scene, it is more public, as anyone can see the flower" (l368 - 370)

- I am not convinced by this explanation, given that anyone can see stars too (if not more so, given their visibility from various vantage points). Because of this I'm also not convinced that the proposed further research will help settle the issue.

[Authors] We agree that the original explanation invoking a private/public distinction based on distance was unconvincing. We have removed it and instead offer a revised interpretation focusing on whether expressions of perceptual availability—such as visible to me or the Japanese verb mieru—typically presuppose the existence of the perceived stimulus. This revision has been incorporated into the manuscript, where we discuss how the difference in ownership ratings between scenes may arise from such presuppositional and pragmatic differences, as follows:

Lines 417-436: We must also consider the dual nature of perceiver-stimulus-prominent sentences. In the star scene, ownership ratings for this sentence type were nearly as high as those for perceiver-prominent sentences, with a significant but small difference. In contrast, in the flower scene, ownership ratings dropped to the level observed for stimulus-only sentences. We propose that this contrast may arise from whether expressions of perceptual availability—such as visible to me or the Japanese verb mieru—typically assume the existence of the perceived stimulus, except in some cases such as pareidolia [27], where one sees a face in the ceiling while knowing that none is actually there. In the star scene, the perceiver-stimulus-prominent sentence tends to presuppose that the star exists and highlights the act of perception (e.g., “The sky was clear, and I was able to see the star”). In some cases, it may also serve to report the discovery of the star’s presence through perception. In the flower scene, however, the same sentence type more naturally evokes a context in which the existence of the flower is not presupposed. It would be pragmatically odd to point to a clearly visible flower in front of both the speaker and the hearer and use this structure to assert its existence. As a result, in the flower scene, the perceiver-stimulus-prominent sentence is more likely to be interpreted similarly to a stimulus-only sentence, which simply reports the presence or state of the object. These results suggest that variation in ownership may reflect whether the sentence foregrounds the perceiver’s experiential access or merely indicates the object’s presence, although further research is needed to determine whether this interpretation is valid. This interpretation aligns with our experimental findings, particularly in the flower scene, where ownership ratings for perceiver-stimulus-prominent sentences were comparable to those for stimulus-only sentences.

7. I am not sure I fully understand the discussion in the paragraph in lines 430 - 438, and I think other readers too might appreciate a little more explanation. What exactly is the claim? Does 'subjective construal of the speaker' here refer to the production of sentences of a certain type or the instantiation of certain mental states? Additionally, is this a causal claim (as implied by 'occur') and, if so, what exactly is causing what?

[Authors] Thank you for the comment. We have re

---

## [Decision Letter · Decision Letter 1]

5 Aug 2025

Sensing minimal self in sentences involving the speaker

PONE-D-25-00361R1

Dear Dr. Imaizumi,

We’re pleased to inform you that your manuscript has been judged scientifically suitable for publication and will be formally accepted for publication once it meets all outstanding technical requirements.

Kind regards,

Laura Morett

Academic Editor

PLOS ONE

Additional Editor Comments (optional):

All comments from the reviewers have been addressed satisfactorily. Therefore, I am pleased to recommend acceptance of this manuscript at this time.

Reviewers' comments:

Reviewer's Responses to Questions

**Comments to the Author**

1. If the authors have adequately addressed your comments raised in a previous round of review and you feel that this manuscript is now acceptable for publication, you may indicate that here to bypass the “Comments to the Author” section, enter your conflict of interest statement in the “Confidential to Editor” section, and submit your "Accept" recommendation.

Reviewer #1: All comments have been addressed

2. Is the manuscript technically sound, and do the data support the conclusions?

Reviewer #1: Yes

3. Has the statistical analysis been performed appropriately and rigorously? 

Reviewer #1: I Don't Know

4. Have the authors made all data underlying the findings in their manuscript fully available?

Reviewer #1: Yes

5. Is the manuscript presented in an intelligible fashion and written in standard English?

Reviewer #1: Yes

6. Review Comments to the Author

Reviewer #1: (No Response)

7. PLOS authors have the option to publish the peer review history of their article (what does this mean?). If published, this will include your full peer review and any attached files.

Reviewer #1: **Yes: **Julian Hauser

---

## [Editor Report · Acceptance letter]

PONE-D-25-00361R1

PLOS ONE

Dear Dr. Imaizumi,

I'm pleased to inform you that your manuscript has been deemed suitable for publication in PLOS ONE. Congratulations! Your manuscript is now being handed over to our production team.

Kind regards,

on behalf of

Dr. Laura Morett

Academic Editor

PLOS ONE